# Antifungal Activity of *Lavandula angustifolia* Essential Oil against *Candida albicans*: Time-Kill Study on Pediatric Sputum Isolates

**DOI:** 10.3390/molecules27196300

**Published:** 2022-09-24

**Authors:** Stefan Mijatovic, Jelena Antic Stankovic, Ivana Colovic Calovski, Eleonora Dubljanin, Dejan Pljevljakusic, Dubravka Bigovic, Aleksandar Dzamic

**Affiliations:** 1Institute of Microbiology and Immunology, Faculty of Medicine, University of Belgrade, 11000 Belgrade, Serbia; 2Department of Microbiology, Faculty of Pharmacy, University of Belgrade, 11000 Belgrade, Serbia; 3Institute for Medicinal Plants Research “Dr Josif Pancic”, 11000 Belgrade, Serbia

**Keywords:** *Candida albicans*, *Lavandula angustifolia*, essential oil, fluconazole, caspofungin, time-kill curve

## Abstract

The aim of our study was to determine the susceptibility of 15 *Candida albicans* sputum isolates on fluconazole and caspofungin, as well as the antifungal potential of *Lavandula angustifolia* essential oil (LAEO). The commercial LAEO was analyzed using gas chromatography-mass spectrometry. The antifungal activity was evaluated using EUCAST protocol. A killing assay was performed to evaluate kinetics of 2% LAEO within 30 min treatment. LAEO with major constituents’ linalool (33.4%) and linalyl acetate (30.5%) effective inhibited grows of *C*. *albicans* in concentration range 0.5–2%. Fluconazole activity was noted in 67% of the isolates with MICs in range 0.06–1 µg/mL. Surprisingly, 40% of isolates were non-wild-type (non-WT), while MICs for WT ranged between 0.125–0.25 µg/mL. There were no significant differences in the LAEO MICs among fluconazole-resistant and fluconazole-susceptible sputum strains (*p* = 0.31) and neither among caspofungin non-WT and WT isolates (*p* = 0.79). The 2% LAEO rapidly achieved 50% growth reduction in all tested strains between 0.2 and 3.5 min. Within 30 min, the same LAEO concentration exhibited a 99.9% reduction in 27% isolates. This study demonstrated that 2% solution of LAEO showed a significant antifungal activity which is equally effective against fluconazole and caspofungin susceptible and less-susceptible strains.

## 1. Introduction

Yeasts of the genus *Candida*, as members of the skin and mucosal microbiota, can cause opportunistic fungal infections, both superficial and invasive. For hospitalized pediatric patients, *Candida* is a significant pathogen identified in the setting of bloodstream infections. Children with acute myeloid leukemia (AML), hematopoietic stem cell transplant (HSCT) recipients and patients admitted to intensive care units (ICU) have the highest risk of invasive candidiasis development [1]. With the risky patients, abnormal *Candida* colonization correlates with a higher probability of its invasive infection [2]. According to the origin, the mentioned mycoses are endogenous, so the damage and the increased permeability of the mucous membranes as consequences of the underlying disease, cytostatic or radiation therapy are the entry points for these yeasts. Studies of the pediatric candidiasis showed that *Candida*
*albicans* was the most common isolated species, but collectively *Candida*
*parapsilosis*, *Candida glabrata*, *Candida lusitaniae* and *Candida krusei* predominated [3,4].

Prophylactic and pre-emptive therapies with high-risk patients are very important in prevention of invasive candidiasis. As drugs with low toxicity, azoles and echinocandins are widely used in prevention and treatment of the fore mentioned mycoses [5]. The introduction of echinocandins into standard protocols alone or in combination with other antifungals has greatly improved the clinical outcomes. Although with lower incidence of adverse events than other antifungals, echinocandins are only available as intravenous formulations [6]. *Candida* resistance to echinocandins has emerged and it is most commonly associated with *C. glabrata* [7]. On the other side, azoles have good tolerability, but a narrow activity spectrum. *C. krusei* is intrinsically resistant to fluconazole whereas *C. glabrata* is highly resistant to the mentioned antifungal [8]. However, the increasing resistance of *C. albicans* and non-albicans species to these antifungals, recurrent infections and persistent colonization require a different strategy in prophylaxis and in treatment of the described disease.

Traditionally, essential oils (EOs) and extracts of aromatic plants have been used in the treatments of wound infections and burns [9,10]. Some EOs in certain cultures are used in aromatherapy as a relaxant and as a carminative and a sedative agent [11]. In addition, dried parts of plants are used as repellents for various arthropods [12]. However, in vitro and clinical studies show different and often conflicting results regarding the efficacy of EOs in human medicine. So far, the antibacterial and antifungal effects of aromatic plants’ extracts and EOs have been investigated. EOs of oregano, thyme and lavender, cloves, cinnamon and tea tree showed broad and potent anti *Candida* activity in in vitro studies [13,14,15]. It is assumed that EOs interfere with fungal membranes, but are distinct from currently available polyenes [16]. Furthermore, inhibition of cell wall polysaccharide synthesis could be another method for EOs antifungal activity [17]. EOs show in vitro activity against *Candida* azole-resistant strains, implicating a different mode of action [18]. It is certain that EOs have an antifungal effect, but there are some limitations. In a laboratory, determinations of effective antimicrobial concentration are difficult to compare because their outcome is influenced primarily by the chemical composition of EOs and methodologies of research. Lipophilic character makes them potentially toxic, especially if used at high concentration and should not be administrated orally and parenterally. Due to their broad-spectrum antimicrobial activity, EOs and their components would be very interesting solution for improving prophylactic and therapeutic failures related to the emergence of resistant strains. Aromatherapy could be an additional method for the prevention of massive upper respiratory tract colonization and the consequent bloodstream, but also some systemic *Candida* infections.

The aim of our study was to determine the susceptibility of *Candida albicans* sputum isolates to *Lavandula angustifolia* essential oil (LAEO) as well as to determine its fungicidal potential on tested cultures in a liquid phase. In addition, we compared the antifungal activity of LAEO to a selected azole (fluconazole) and a representative echinocandin (caspofungin).

## 2. Results

### 2.1. Microbial Identification

Out of the 15 analyzed *Candida* isolates, 10 strains were recognized as *C. albicans* using chromogenic media. Additional five isolates were identified by matrix-assisted laser desorption in ionization-time of flight mass spectrometry (MALDI-TOF MS). Peptide mass fingerprinting of these isolates as well as *C. albicans* ATCC 10259 performed by MALDI-TOF MS are shown in Figure 1.

### 2.2. Chemical Composition of Essential Oil

A total of 50 compounds were detected in LAEO (Figure 2), of which 30 were identified (97.4% of whole composition) (Table 1). These constituents could be classified in six different groups: monoterpene hydrocarbons, oxygenated monoterpenes, sesquiterpene hydrocarbons, oxygenated sesquiterpenes, diterpene hydrocarbons and oxygenated diterpenes. In Table 1, the 10 main components determined by European Pharmacopoeia representing 84.6% of the whole LAEO composition are bolded [19]. The main constituents of the LAEO were linalool and linalyl acetate, constituting 33.4% and 30.5%, respectively. The contribution of 1,8-cineole (5.0%) and camphor (4.6%) deviated from the ranges determined by European Pharmacopoeia and the ISO 3515 standard, therefore this LAEO can be considered an essential oil of non-standard quality.

### 2.3. Minimum Inhibitory Concentration (MIC) and Minimum Fungicidal Concentration (MFC)

The MICs of LAEO that effectively inhibited grows of *C*. *albicans* sputum strains ranged from 0.5 to 2% (mean MIC 1.17%; MIC_50_ 1%; MIC_90_ 2%) (Table 2). LAEO MFCs for the tested isolates ranged between 1–4% (mean MFC 1.93%; MFC_50_ 2%; MFC_90_ 2%). The MIC and MFC of LAEO for *C. albicans* ATCC 10259 were in the range observed among clinical isolates (MIC 2%; MFC 2%).

Five clinical strains (*C. albicans* PED01, PED03, PED07, PED08 and PED09) were resistant to fluconazole with MICs 64 µg/mL, whereas MICs for susceptible isolates ranged from 0.06 to 1 µg/mL (mean MIC 21.57 µg/mL; MIC_50_ 0.5 µg/mL; MIC_90_ 64 µg/mL). Fluconazole MFC_50_ and MFC_90_ were >64 µg/mL. Six sputum isolates (*C*. *albicans* PED03, PED07, PED08, PED09, PED12 and PED15) were categorized as caspofungin non-WT strains with MICs > 0.25 µg/mL, while MIC for WT isolates ranged between 0.125–0.25 µg/mL (mean MIC 0.57 µg/mL; MIC_50_ 0.25 µg/mL; MIC_90_ 0.5 µg/mL). Caspofungin concentration 2 µg/mL completely kills 50% and 90% of tested isolates. In addition, there were no significant differences in the LAEO MICs among fluconazole-resistant (median 1%) and fluconazole-susceptible (median 1%) sputum strains (Mann–Whitney U test; *p* = 0.31). In addition, no LAEO MICs differences were observed among caspofungin non-WT (median 1%) and WT (median 1%) isolates (Mann–Whitney U test; *p* = 0.79).

### 2.4. Time-Kill Assay

The time-kill curves of the 2% LAEO against the 16 *Candida* strains are illustrated in Figure 3. Time intervals to reach 50%, 90% and 99.9% reduction in the number of yeast cells from the starting inoculums for each strain are summarized in Table 3. After 30 min exposure to LAEO in 4 strains, the cell count was reduced ≥3.0 log_10_ CFU/mL in range from 3.0 to 3.2 log_10_ CFU/mL, whereas in the remaining 12 strains, the reduction ranged only 0.47 to 2.47 log_10_ CFU/mL. This implicates that the mentioned concentration of LAEO showed more fungistatic than fungicidal activity during 30 min exposure.

A 50% growth reduction for all sputum strains was rapidly achieved between 0.2 and 3.5 min (approximately 0.83 min). In addition, the time points required to reach 90% and 99.9% were more variable among the tested strains. In fact, within 30 min of exposure to LAEO, 90% reduction was observed with nine strains (*C*. *albicans* PED01, PED03, PED05, PED06, PED07, PED08, PED09, PED13 and PED14) in time range 30 s–18 min. LAEO exhibited 99.9% reduction in four strains (*C*. *albicans* PED01, PED03, PED05 and PED06) between 1.8 and 29.7 min. For other isolates, together with *C. albicans* ATCC 10259 no reduction was achieved in the CFU of 90% and 99.9% during 30 min of LAEO exposure.

The number of *C*. *albicans* cells significantly declined over time in all LAEO treated sputum strains (Friedman test; *p* < 0.05) (Figure 4). Pairwise comparisons using Wilcoxon signed rank test for related samples showed significant differences in all measurement times except 5 min–15 min (*p* = 0.14) and 15 min–30 min (*p* = 0.06) (Figure 4).

Among sputum strains 2% LAEO was reducing the cell counts rapidly after 1 min by approximately 0.49 log_10_ CFU/mL (68%) (Figure 5). This activity remained, reducing cell viability by approximately 0.73 log_10_ CFU/mL (81%), 0.85 log_10_ CFU/mL (86%), 0.91 log_10_ CFU/mL (87%) and 1.03 log_10_ CFU/mL (91%) within 2, 5, 15 and 30 min, respectively (Figure 5). In *C. albicans* ATCC 10259 cells reduction calculated at 2, 5, 15 and 30 min were 0.41 log_10_ CFU/mL (60%), 0.48 log_10_ CFU/mL (67%), 0.77 log_10_ CFU/mL (83%), 0.77 log_10_ CFU/mL (83%) and 0.91 log_10_ CFU/mL (88%), respectively (Figure 5).

## 3. Discussion

As natural aromatic plant products, EOs are mainly composed of terpens and terpenoids. LAEO is present with approximately 1.3% in the flowering aerial part of the plant and contains more than 100 compounds [20]. It was shown that the LAEO and its main constituents can kill or inhibit the growth of fungi [21]. The main components of this LAEO that exhibit fungistatic and fungicidal effects are monoterpene alcohols (60–65%), such as linalool (20–50% per fraction) and linalyl acetate (25–46% per fraction) [20]. The other components are cis-ocimene, terpinen4-ol, limonene, cineol, camphor, lavandulyl acetate, lavandulol and α-terpineol, β-caryophyllene, geraniol, α-pinene. Non-terpenic aliphatic components such as 3-octanon, 1-octen-3-ol, 1-octen-3-y-lacetate, 3-octanol are present as well [22]. They are determined based on the chemical compositions LAEO, but EOs components could differ among other *Lavandula* spp. [23]. A chemical analysis of *L. binaludensis* EO showed that γ-terpinen with participation of 71.6% is its major content [24]. Zuzarte et al. showed that the main component of *L. viridis* EO was 1,8-cineole with participation of 34.3–42.2% [25]. Out of 10 lavender EO components shown in Table 1, 8 are in range classified by the European Pharmacopoeia [19]. Two components of 1,8-cineole and camphor were in higher percentage (5.0% and 4.6%, respectively). These deviations in EOs’ composition sometimes reflect non-uniformity in the bioactivity results and they could be consequences of climatic, geographical, seasonal and geological factors, physiological characteristics of plants and extraction procedures.

The action mechanism of essential oils remains somewhat controversial. An exceptional antifungal activity of EOs is probably based on their lypophilic nature which typically integrates into membrane structures thus increasing the cell permeability and leaching of intracellular membranes [26]. It has been noticed that among lavender EOs, the component α-pinene shows the best antifungal effect [25]. Other components such as linalool 1,8-cineol and linalyl acetate are also potent, contributing to the overall antifungal activity of [27,28]. In addition, linalool decreases *C. albicans* yeast cells rapidly, as the time kill essays show [21]. As an acyclic monoterpene alcohol, linalool differs from most of the monoterpenes that exhibit significant antifungal activity. It is reasonable to speculate that the presence of alcohol components primarily determines the antifungal activity and then whether the constituent has a cyclic or acyclic structure. Probably these terpene alcohol moieties disrupt the microbial cell membrane thus increasing the permeability to protons and larger ions [29].

In this investigation, LAEO exhibited considerable antifungal effects against *C. albicans* sputum isolates. Our results correlate with those of several previous studies in which MIC values for LAEO range between 0.5–2% [30,31]. However, LAEO MIC_50_ and MIC_90_ values were twice higher than the values obtained in the previous experiments. This may be a consequence of the denser inocula that we used, as well as the origin of the tested strains. D’Auria et al. showed that higher concentrations of LAEO are necessary for growth inhibition of *C. albicans* oropharingeal isolates [21]. Moreover, a small number of studies included the mentioned isolates in their research, meaning that we could thoroughly compare our results with them. Nevertheless, the authors who compared *Candida* susceptibility to EOs by vapor diffusion and microdilution reported that MIC values in the vapor phase were lower than in the liquid phase [32,33]. Mandras et al. obtained an LAEO MIC_90_ of 1% in the liquid phase; whereas in the vapor phase, the growth of all *C. albicans* strains tested at a 0.06% concentration was inhibited [34]. High volatility, poor water solubility and dispersibility of EOs and their components could be the main reasons for unequal inter-laboratory results.

The experiments performed in our study showed that LAEO MICs of the all-tested fluconazole-resistant *C. albicans* strains were not different from the MICs of fluconazole-susceptible strains implicating that *C. albicans* does not exhibit cross-resistance to the mentioned EO and fluconazole. This could be a consequence of different antifungal actions in which azoles interfere with cytochrome P-450-dependent enzyme lanosterol 14α-demethylase, which is responsible for the production of ergosterol and where several mechanisms of fluconazole resistance are possible: a decrease in the affinity of lanosterol 14α-demethylase, hyperproduction of the same enzyme, as well as a loss of sterol desaturase, leading to alteration in the ergosterol synthesis [8]. Moreover, EOs can disrupt the function of efflux pumps which, through an active transfer, scavenge the fluconazole across the plasma membrane allowing its easier entry inside the yeast cell [35]. Some authors implicate that EOs alter the synthesis of many cell wall polysaccharide components, such as β-glucan, chitin and mannan [17]. Surprisingly, our data have shown that strains in which the caspofungin MIC values were higher than 0.25 µg/mL did not require higher LAEO concentrations for their growth inhibition.

Grounded on the fact that MFC values were once or twice higher than MICs (Table 1), it is suggested that LAEO has more fungicidal than fungistatic activity during 24 h exposure [36]. Based on the MFC found for each *Candida* species, time-kill curves were constructed in order to evaluate the effect of MFC_90_ LAEO concentration during short time exposure. The kill curves obtained with 2% LAEO indicate that this concentration is primary fungistatic (< 3 log_10_ CFU/mL) against most *C. albicans* isolates for 30 min. Previous studies have shown that during a short time exposure, 2% concentration of EOs exhibit a predominantly fungistatic activity against *Candida*, but fungicidal activity may also be observed, depending on the strain, type of EO and the test conditions [21,37]. However, in our investigation, the aforementioned LAEO concentration led to 0.301 log_10_ CFU/mL (50%) yeast cells reduction within 5 min for all the tested strains. The fact that 99.9% of growth reduction was not achieved for most isolates during 30 min implies that the decreasing of the significant reduction effect observed in the first minutes of the exposure is probably in relation to fungal cell stress adaptation. This characteristic was also observed during exposure of EOs and their components to yeast cells where many stress responses and signaling pathways were induced, such as oxygen and free radical detoxification, heat shock proteins and chaperones, autophagy and vacuolar degradation mechanisms [38]. The same study showed that these responses are arising from early 15 min to late 90 min of exposure to certain EOs components.

Regarding the antifungal efficiency of the 2% LAEO against *C. albicans* strains, our EO was able to significantly reduce the number of yeast cells during a short time exposure. Unfortunately, only one study monitored the effect of LAEO against *Candida* during 30 min of exposure [21]. In the same study, 2% LAEO concentration killed 99% of the cells within 5 min and 100% of the cells were killed within 15 min. This study was performed on one strain only, so it is difficult to compare the results obtained then with the ones in our investigation. However, in our investigation the mean cell reduction calculated in 5 min and 15 min for sputum strains was approximately 86% and 88%, respectively. Similarly, cell reduction in *C. albicans* ATCC 10259 calculated within 5 min and 15 min was 83%.

## 4. Materials and Methods

### 4.1. Microorganisms

Fifteen species of *Candida albicans* isolated from sputum of pediatric patients with AML, diagnosed at the Institute for Mother and Child Health Care of Serbia “Dr Vukan Cupic”, were included in this study. Initially, isolation was performed on Sabouraud dextrose agar (SDA) (HiMedia, Mumbai, India). Identification of *Candida* strains was performed after subcultivation on HiCromeTM Candida Differential Agar (HiMedia, Mumbai, India) according to the specific appearance of colonies defined by the manufacturer. For isolates that could not be identified by the aforementioned chromogenic medium, protein fingerprint analysis obtained by matrix-assisted laser desorption in ionization-time of flight mass spectrometry (MALDI-TOF MS) was used. After subcultivation on SDA at 37 °C for 24 h colony adhering to the toothpick was directly transferred onto a 96-spot polished steel target plate (Bruker Daltonics, Bremen, Germany) for spotting and allowed to air drying. The bacterial test standard was used as a positive control (Bruker Daltonics, Bremen, Germany). Immediately, 1 µL of 70% formic acid and alpha-cyano-4-hydroxycinnamic acid (HCCA) matrix (Bruker Daltonics, Bremen, Germany) were spotted onto the target plate. After drying at room temperature samples were analyzed using MALDI-TOF MS Biotyper Sirius one IVD System (Bruker Daltonics, Bremen, Germany) in automatic runs operated by flexControl, ver. 3.4.207.20 (Bruker Daltonics, Bremen, Germany). The yeast identification was achieved using MBT Compass software, ver. 4.1.100 (Bruker Daltonics, Bremen, Germany), based on the comparison of generated mass spectra with database (MBT Compass Library, Revision H, 3893 species/entries). The log score values > 1.7 indicated reliable species identification. American Type Culture Collection strain *C. albicans* ATCC 10259 as control strain suggested for EOs assays was included in the investigation as well. Complete mycological examinations were performed at the National Referential Medical Mycology Laboratory (NRMML), Faculty of Medicine, University of Belgrade (Serbia).

### 4.2. Essential Oil

*L.**angustifolia* essential oil was obtained from Herba^®^ (Belgrade, Serbia) lot N° 02560115 and stored at −18 °C until the chemical and mycological investigations were to be conducted.

### 4.3. Chemical Identification of Essential Oil

The gas chromatography with flame ionization detection (GC-FID) analyses were carried out with a HP-7890 Series A apparatus (Agilent Technologies Inc., Santa Clara, CA, USA) equipped with a split-splitless injector, a flame ionization detector (FID), and a HP-5 capillary column (25 m × 0.32 mm i.d., film thickness 0.52 mm). The oven temperature was programmed rising from 40 to 260 °C at 4 °C/min; injector temperature was 250 °C; detector temperature was 300 °C; carrier gas was H2 (1.0 mL/min). The relative contents expressed as percentages were obtained from electronic integration of the peak areas measured using FID.

The gas chromatography-mass spectrometry (GC/MS) analyses were performed under almost the same analytical conditions as the GC-FID analyses, with a HP G1800C Series II GCD analytical system (HewlettPackard, Palo Alto, CA, USA) equipped with a HP-5MS capillary column (30 m × 0.25 mm i.d., film thickness 0.25 mm). Helium (1.0 mL/min) was used as carrier gas, and the transfer-line temperature (MSD) was heated at 260 °C. Mass spectra were acquired in the EI mode (70 eV) over the m/z range 40–450 amu. Aliquot of 1 µL of sample diluted in EtOH (10 mL/mL) was injected in split mode (1:30). The identification of the constituents was performed by comparing their mass spectra and retention indices (RIs) with those obtained from authentic samples and/or 4 listed in the NIST/Wiley mass-spectra libraries, using different types of searches (PBM/NIST/AMDIS) and available literature data [39,40].

### 4.4. Antifungal Agents

Fluconazole and caspofungin (Sigma–Aldrich, Taufkirchen, Germany) powders were dissolved in 100% dimethyl sulfoxide (DMSO) at final concentration 12.8 mg/mL and 0.8 mg/mL, respectively. Prepared stock solutions were stored at −70 °C until use.

### 4.5. Antifungal Susceptibility Testing

#### 4.5.1. Preparation of Inoculums

Different colonies of 15 *Candida albicans* isolates and a control strain from overnight culture on SDA at 37 °C were suspended in 6 mL distilled water and homogenized with a gyratory vortex to the final density of the 0.5 McFarland standard. Working suspension was made by further 1:10 dilution of the standardized suspension resulting in 1–5 × 10^5^ CFU/mL. Cell counts and viability of yeasts were confirmed in triplicates using a hemocytometer chamber by adding trypan blue.

#### 4.5.2. Determination of LAEO, Fluconazole and Caspofungin MIC and MFC Values

LAEO stock solution was prepared in pure ethanol (1:2) and then diluted in RPMI 1640 broth with L-glutamine (Capricorn Scientific, Ebsdorfergrund, Germany), buffered to pH 7.0 with 0.165 M morpholinepropanesulfonic acid (MOPS; Sigma-Aldrich, Taufkirchen, Germany), supplemented with 2% glucose to obtain its final concentration of 8% (*vol./vol.*). Tween 80 in final concentration of 0.5% was added to enhance the EO solubility. Initially, 200 µL of freshly prepared LAEO stock solution was poured in the first rows of flat-bottomed 96-well microtiter plates. One hundred µL of RPMI 1640 with MOPS and 2% glucose were added in the remained wells. Nine serial two-fold dilutions were performed by the withdrawal of a 100 µL aliquot from the concentrated well into the succeeding one. Preparations of fluconazole and caspofungin work solutions, together with the susceptibility testing, were performed according to the EUCAST (European Committee on Antimicrobial Susceptibility Testing) document E.DEF 7.3.2. After adding of 100 µL appropriate working yeast suspension (1–5 × 10^5^ CFU/mL), the final concentration of LAEO ranging between 4–0.008%, 64–0.125 µg/mL for fluconazole and 4–0.008 µg/mL for caspofungin were obtained. EO-free and antifungals-free grow control and sterility control wells were also included. The final solvent concentrations did not affect yeast growth as their concentration did not exceed 2% for ethanol and 1% for DMSO. All microtiter plates were incubated at 37 °C for 24 h in aerobic atmosphere.

After incubation, the plates were estimated to fungal growth. Minimal inhibitory concentration (MIC) of LAEO was defined as the lowest concentration, showing complete inhibition of visible growth. For fluconazole and caspofungin, MIC determinated the lowest concentration reducing ≥ 50% of the culture growth in comparison with the drug-free growth control reading by the microplate reader on 450 nm (Tecan Sunrise, Mannedorf, Switzerland). EUCAST clinical breakpoints are used to categorize the fluconazole susceptibility. Due to the significant inter-laboratory variation in caspofungin MIC ranges, a recommended epidemiological cut-off (ECV) value of 0.25 µg/mL was used to categorize the isolates into wild-type (WT) and non-WT [41]. MIC_50_ and MIC_90_ were defined as the lowest concentrations of each substance capable of inhibiting 50% and 90% of the tested isolates, respectively.

The minimal fungicidal concentration (MFC) of EO and tested drugs were estimated after MIC determination as the lowest concentration that kills 99.9% yeast cells after subculturing 10 μL of broth taken from all the wells without turbidity on SDA. MFC_50_ and MFC_90_ were defined as the lowest concentrations that kill 50% and 90% of tested isolates, respectively. Each experiment was performed in duplicates on three different dates. The results were reported as modal values.

### 4.6. Time-Kill Essay

In order to evaluate the further fungicidal effect of LAEO MFC_90_, a time-kill assay on 15 *C. albicans* sputum isolates and *C. albicans* ATCC 10259 was performed in a liquid phase according to the method proposed by Klepser et al. with some modifications [42]. One hundred microliters of a working suspension 1–5 × 10^5^ CFU/mL was poured onto flat-bottomed 96-well micro titration plates and 100 µL of 4% EO, previously prepared by dissolving in pure ethanol (1:2) and further diluting in RPMI 1640 broth with MOPS, 2% glucose and 0.5% Tween 80, were added (the final EO concentration was 2%). Comparatively, nontreated growth controls were prepared as well. After incubation at 0, 1, 2, 5, 15 and 30 min, 10 µL from each test samples were removed for three serial 1:10 dilutions (10 µL samples and 90 µL distilled water). Colonies were counted after 100 µL inoculation of the second and the third dilution on SDA and incubation at 37 °C for 48 h in aerobic atmosphere. The limit of detection was ≥ 10^2^ CFU/mL. The experiment was performed in duplicates and average values are included for further observations. Time-kill curves were constructed by plotting logarithmic values of colony counts (log_10_ CFU/mL) as a function of time (in minutes). In order to determine fungicidal or fungistatic character of LAEO, a reduction in the growth ≥ 3 log_10_ CFU/mL of the starting inoculums was defined as a fungicidal activity of EO [43]. Approximate time intervals to achieve 50% (≥0.301 log_10_ CFU/mL), 90% (≥1 log_10_ CFU/mL) and 99.9% (≥3 log_10_ CFU/mL) of growth reduction were calculated with non-linear interpolation function using Excel (version 16; Microsoft Corp., Redmond, WA, USA) for Windows 10 (Microsoft Corp., Redmond, WA, USA). Ultimately, roughly counted, the overall reduction in *C. albicans* sputum strains for each time interval expressed as a log_10_ CFU/mL was calculated using the following Equation (1):(1)$$mean\;\log_{10}\mathrm{CFU}/\mathrm{mL}\;reduction=-\log_{10}\left(1-\frac{mean\;\%\;reduction}{100}\right)$$

### 4.7. Statistical Analysis

Normality distribution of the variables was verified using the Kolmogorov–Smirnov and Shapiro–Wilk tests. MIC values were compared by using the Mann–Whitney U test. For comparison of the time-kill test results Friedman test with post hoc pairwise comparison (Wilcoxon signed-rang test) with applied Bonferroni correction were used to determine the correlation between log_10_ CFU/mL values measured at different times. In all the cases, *p* values < 0.05 were considered as statistically significant. All statistical analyses were performed using Excel (version 16; Microsoft Corp., Redmond, WA, USA) for Windows 10 (Microsoft Corp., Redmond, WA, USA) and R Commander (R package ver. 2.7-1; R Core Team, Vienna, Austria).

## 5. Conclusions

As expected, the major components of LAEO were linalool and linalyl acetate. Our results demonstrated the efficacy of LAEO in growth reduction in *C. albicans* sputum isolates. The 2% solutions of LAEO possess potent and rapid antifungal activity that is primary fungistatic within 30 min. The same concentration is equally effective against fluconazole susceptible and fluconazole resistant, as well as against WT and non-WT caspofungin isolates. Although the prophylactic treatment of invasive candidiasis is based on systemic antifungal agents, further studies are necessary to assess the application of the essential oils as aromatherapy with high-risk patients.

## Figures and Tables

**Figure 1 molecules-27-06300-f001:**
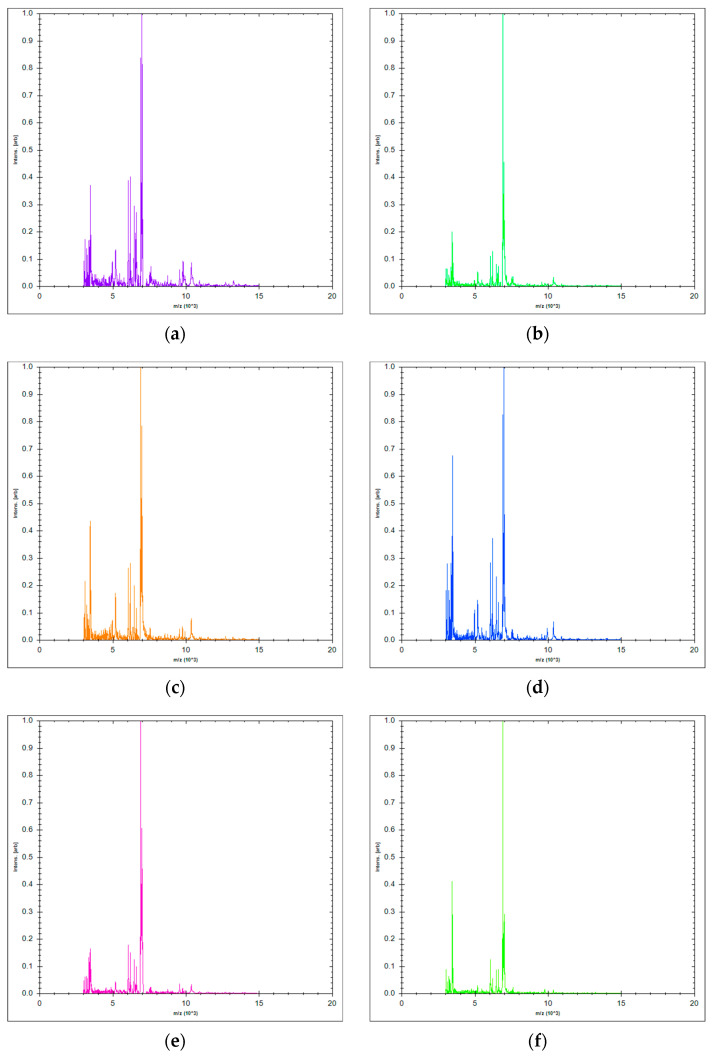
MALDI-TOF mass spectra (*m*/*z* 3000 to 15,000) of: (**a**) *C. albicans* PED02; (**b**) *C. albicans* PED06; (**c**) *C. albicans* PED12; (**d**) *C. albicans* PED13; (**e**) *C. albicans* PED15; (**f**) *C. albicans* ATCC 10259.

**Figure 2 molecules-27-06300-f002:**
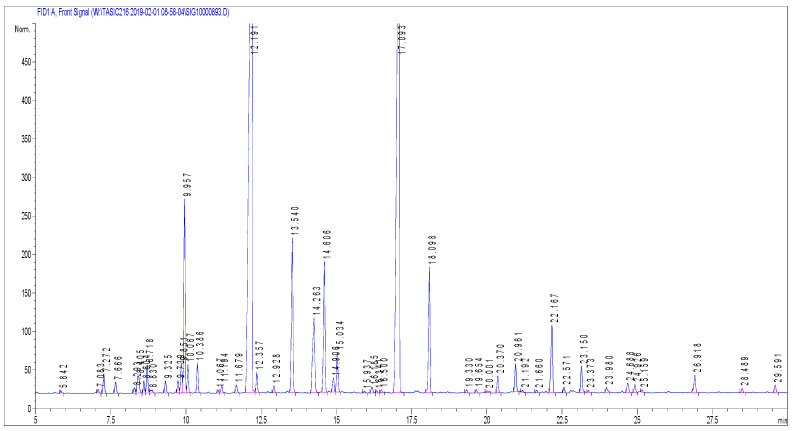
Chromatogram of *L. angustifolia* essential oil showing the separation of 50 chemical components.

**Figure 3 molecules-27-06300-f003:**
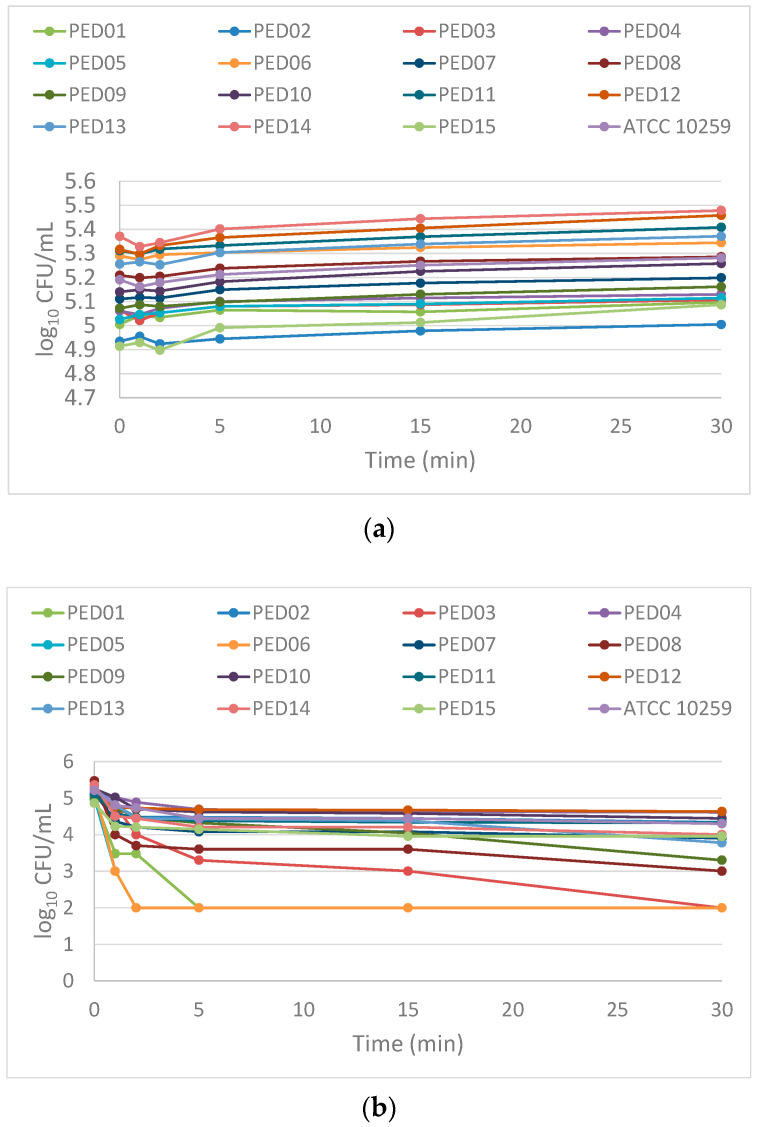
Time-kill of *C. albicans* sputum strains on different times (0–30 min): (**a**) untreated (grow controls) of *C. albicans* (PED01–PED15) and *C. albicans* ATCC 10259; (**b**) *C. albicans* isolates (PED01–PED15) and *C. albicans* ATCC 10259 treated with 2% concentration of lavender essential oil (LAEO); Within 30 min exposure in 4 strains (PED01, PED03, PED05, PED06) cell count was reduced ≥ 3.0 log_10_ CFU/mL, where in rest 12 strains reduction ranged from 0.47 log_10_ CFU/mL (PED04) to 2.47 log_10_ CFU/mL (PED08).

**Figure 4 molecules-27-06300-f004:**
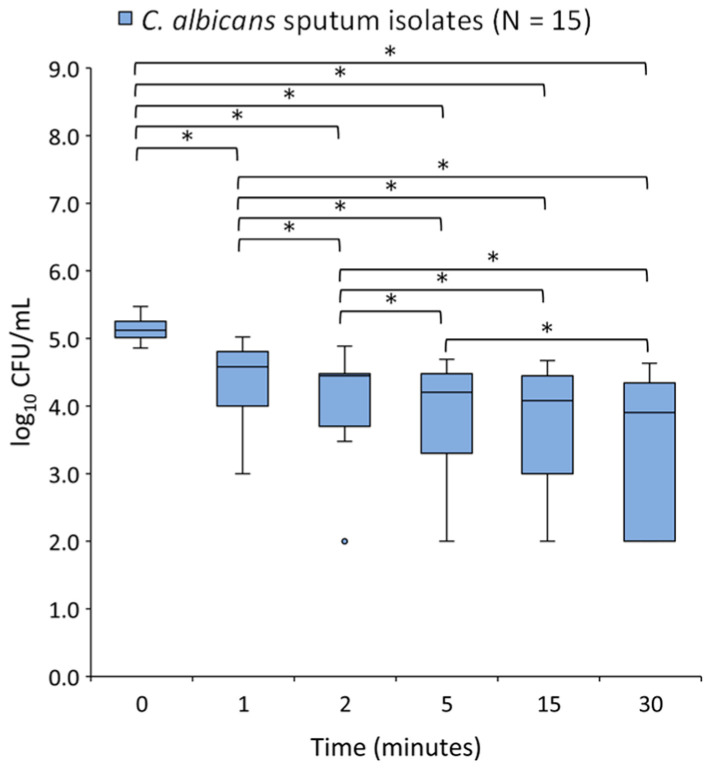
Box plot of cell counts (log_10_ CFU/mL) of 15 *C. albicans* sputum strains noted from pre-exposure (0 min) and within 30 min exposure to 2% lavender essential oil (LAEO). Friedman test with post hoc pairwise comparison with applied Bonferroni correction showed significant differences among all pairs (* *p* < 0.05) except 5 min–15 min and 15 min–30 min.

**Figure 5 molecules-27-06300-f005:**
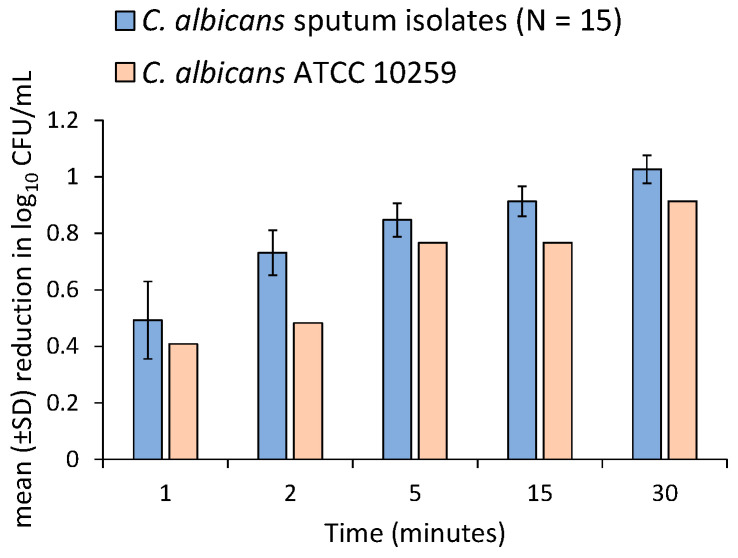
Bar chart of mean (±SD) reduction in log_10_ CFU/mL levels of 15 *C. albicans* sputum isolates as well as *C. albicans* ATCC 10259 from 1 to 30 min exposure to 2% of *L.*
*angustifolia* essential oil. Within 1 min treatment mean cell reduction in sputum strains was 0.49 log_10_ CFU/mL (68%), while in 30 min their cell viability mean reduced 1.03 log_10_ CFU/mL (91%). In *C. albicans* ATCC 10259 cell count reduction ranged from 0.41 log_10_ CFU/mL (61%) to 0.91 log_10_ CFU/mL (88%) calculated in 1 min and 30 min, respectively.

**Table 1 molecules-27-06300-t001:** Composition of the *L. angustifolia* essential oil used in this study.

Components	Contents (%) According to Ph. Eur ^1^	Content (%) of LAEO ^2^ Sample
α-thujene		0.1
α-pinene		0.5
camphene		0.3
octen-3-ol		0.2
**octanone-3**	0.1–5	0.8
myrcene		0.7
α-phellandrene		0.3
p-cymene		0.3
**limonene**	Max 1	0.9
**1,8-cineole**	Max 2.5	5.0
cis-β-ocimene		0.7
trans-β-ocimene		0.7
linalool oxide cis		0.2
linalool oxide trans		0.2
**linalool**	20–45	33.4
octenyl acetate		0.5
**camphor**	Max 1.2	4.6
borneol		3.2
**terpinene-4-ol**	0.1–8	3.9
**lavandulol**	Min 0.1	0.4
**α-terpineol**	Max 2	1.2
hexyl isovalerate		0.2
**linalyl acetate**	25–47	30.5
**lavandulyl acetate**	Min 0.2	3.5
neryl acetate		0.5
geranyl acetate		0.8
β-caryophyllene		2.3
trans-α-bergamotene		0.2
trans-β-farnesene		0.8
caryophyllene oxide		0.6

^1^ European Pharmacopoeia. ^2^
*L. angustifolia* essential oil.

**Table 2 molecules-27-06300-t002:** MIC and MFC values of *L. angustifolia* essential oil and drugs against *Candida albicans* strains evaluated by the broth microdilution method.

Strains	LAEO ^1^		Fluconazole		Caspofungin	
	MIC ^2^	MFC ^3^	MIC	MFC	MIC	MFC
*C. albicans* PED01	1	2	64	>64	0.25	2
*C. albicans* PED02	1	2	0.5	>64	0.25	2
*C. albicans* PED03	0.5	1	64	>64	4	4
*C. albicans* PED04	1	2	1	>64	0.25	0.5
*C. albicans* PED05	1	2	0.5	>64	0.25	0.25
*C. albicans* PED06	2	2	0.125	>64	0.25	2
*C. albicans* PED07	1	2	64	>64	0.5	0.5
*C. albicans* PED08	1	2	64	>64	0.5	2
*C. albicans* PED09	1	2	64	>64	0.5	2
*C. albicans* PED10	0.5	1	0.06	>64	0.25	1
*C. albicans* PED11	0.5	1	0.125	>64	0.25	2
*C. albicans* PED12	2	4	0.5	>64	0.5	0.5
*C. albicans* PED13	2	2	0.5	>64	0.25	2
*C. albicans* PED14	2	2	0.125	>64	0.125	0.5
*C. albicans* PED15	1	2	0.125	>64	0.5	1
*C. albicans* ATCC 10259	2	2	0.5	>64	0.5	2

^1^ LAEO = *L. angustifolia* essential oil; ^2^ MIC = minimum inhibitory concentration (values expressed in %vol./vol.); MIC for fluconazole and caspofunin are expressed in µg/mL; ^3^ MFC = minimum fungicidal concentration (values expressed in %vol./vol.); MFC for fluconazole and caspofunin are expressed in µg/mL.

**Table 3 molecules-27-06300-t003:** Times for 2% *L. angustifolia* essential oil to achieve 50%, 90% and 99.9% reductions in growth of starting inoculum during 30 min treatment.

Strains	2% Solution of LAEO ^1^		
	50% Reduction ^2^	90% Reduction	99.9% Reduction
*C. albicans* PED01	0.2	0.7	5
*C. albicans* PED02	1.4	>30	>30
*C. albicans* PED03	1.1	2	29.7
*C. albicans* PED04	3.5	>30	>30
*C. albicans* PED05	0.2	0.6	2
*C. albicans* PED06	0.2	0.5	1.8
*C. albicans* PED07	0.4	3.3	>30
*C. albicans* PED08	0.2	0.7	>30
*C. albicans* PED09	1.1	15.2	>30
*C. albicans* PED10	1.3	>30	>30
*C. albicans* PED11	0.6	>30	>30
*C. albicans* PED12	0.6	>30	>30
*C. albicans* PED13	0.7	18	>30
*C. albicans* PED14	0.4	3.1	>30
*C. albicans* PED15	0.5	>30	>30
*C. albicans* ATCC 10259	0.8	>30	>30

^1^ LAEO = *L. angustifolia* essential oil; ^2^ Values are expressed in minutes.

## Data Availability

Not applicable.

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
