# Peer review of "Antifungal Activity of *Lavandula angustifolia* Essential Oil against *Candida albicans*: Time-Kill Study on Pediatric Sputum Isolates"

_molecules, 2022, doi:10.3390/molecules27196300_

Round 1
Reviewer 1 Report
The study deals with the current and interesting topic of the antimicrobial effect of lavender essential oil on Candida albicans isolates. Overall, I consider the study to be well written with well drawn conclusions. Unfortunately, quite a few C. albicans isolates were included in the study for the possibility of greater generalization of the results using MIC50 and MIC90.
I have a few small comments about the publication:
1/ It is necessary to go through the text and correctly set the writing of the text in italics, e.g. missing italics on L. 64, etc.
2/ The evaluation of MIC50 and MIC90 for 15 isolates is quite a small number of strains for a larger indicative value (let's say the absolute minimum). However, data interpreted in this way also appear in the literature.
3/ L. 100 redundant dot
4/ Try to write the Latin words of microorganisms in Figures in italics.
Author Response
Dear reviewer,
Thanks for the comments.
We send our answers:
1. Term in vitro is corrected as in vitro (previous L. 64, now L. 65).
2. Yes, this is a good observation. We have cited similar manuscripts.
3. Redundant dot in text (previous L. 100, now L. 112) – corrected.
4. In Figure 4 and Figure 5 latin names of microorganisms are corrected in italics.
Reviewer 2 Report
Dear Authors,
thank you for your work and results. All of my comments and questions you'll find in text.
best Regards,
MK

Author Response
Response to Reviewer 2 Comments
Dear reviewer,
Thanks for the helpful suggestions
We are sending corrections:
1.L. 51 (previous L. 50): term Candida’s corrected as Candida.
2. L. 52 (previous L. 51): first time we use full name Candida glabrata on L. 43 – corrected.
3. Second and next time short name L. angustifolia – corrected.
4. We added Figure (Figure 2) which represents a chromatogram of 50 detected components of L. angustifolia (Page 4); In Table 1 we inserted all of components which are identified (Page 4 and Page 5).
5. The Department of Parasitology and Mycology of the Institute of Microbiology and Immunology, Faculty of Medicine University of Belgrade has a legal contract for the routine laboratory diagnosis of fungal infections of AML and ALL patients cured in Institute for Mother and Child Health Care of Serbia “Dr Vukan Cupic“. This includes the isolation of Candida and other fungal agents from the sputum sample.
6. The methodology for Candida identification by MALDI-TOF MS is added (Page 10; L. 280 – L. 294); Results obtained by aforementioned method are aded in section 2.1. (Page 2; L. 87 – L. 90) and presented in Figure 1 (Page 3).
Reviewer 3 Report
Dear authors, there are some minor correction and showed on the manuscript.

Author Response
Response to Reviewer 3 Comments
Dear reviewer,
Thanks for the comments
We send our answers:
1. ml and μl are corrected as mL and μL in all manuscript.
2. Term in vitro is corrected as in vitro (previous L. 64, now L. 65).
3. The preparation of L. angustifolia essention oil for time-kill assay is explained in section 4.6. (Page 12; L. 375 – L. 377).

Round 2
Reviewer 2 Report
Dear Authors,
thanky you for your work and added my opinion.
best Regards,
MK